# Effects of Acute Hypoxia on Heart Rate Variability in Patients with Pulmonary Vascular Disease

**DOI:** 10.3390/jcm12051782

**Published:** 2023-02-23

**Authors:** Martina Meszaros, Simon R. Schneider, Laura C. Mayer, Mona Lichtblau, Martino F. Pengo, Charlotte Berlier, Stéphanie Saxer, Michael Furian, Konrad E. Bloch, Silvia Ulrich, Esther I. Schwarz

**Affiliations:** 1Department of Pulmonology, University Hospital of Zurich, 8091 Zurich, Switzerland; 2Department of Health Sciences and Medicine, University of Lucerne, 6002 Lucerne, Switzerland; 3Istituto Auxologico Italiano IRCCS, Department of Cardiology, San Luca Hospital, 20149 Milan, Italy; 4School of Medicine and Surgery, University of Milano-Bicocca, 20122 Milan, Italy; 5Medical Faculty, University of Zurich, 8006 Zurich, Switzerland

**Keywords:** heart rate variability, HRV, pulmonary vascular disease, PVD, pulmonary arterial hypertension, PAH, chronic thromboembolic pulmonary hypertension, CTEPH, hypoxia

## Abstract

Pulmonary vascular diseases (PVDs), defined as arterial or chronic thromboembolic pulmonary hypertension, are associated with autonomic cardiovascular dysregulation. Resting heart rate variability (HRV) is commonly used to assess autonomic function. Hypoxia is associated with sympathetic overactivation and patients with PVD might be particularly vulnerable to hypoxia-induced autonomic dysregulation. In a randomised crossover trial, 17 stable patients with PVD (resting PaO_2_ ≥ 7.3 kPa) were exposed to ambient air (FiO_2_ = 21%) and normobaric hypoxia (FiO_2_ = 15%) in random order. Indices of resting HRV were derived from two nonoverlapping 5–10-min three-lead electrocardiography segments. We found a significant increase in all time- and frequency-domain HRV measures in response to normobaric hypoxia. There was a significant increase in root mean squared sum difference of RR intervals (RMSSD; 33.49 (27.14) vs. 20.76 (25.19) ms; *p* < 0.01) and RR50 count divided by the total number of all RR intervals (pRR50; 2.75 (7.81) vs. 2.24 (3.39) ms; *p* = 0.03) values in normobaric hypoxia compared to ambient air. Both high-frequency (HF; 431.40 (661.56) vs. 183.70 (251.25) ms^2^; *p* < 0.01) and low-frequency (LF; 558.60 (746.10) vs. 203.90 (425.63) ms^2^; *p* = 0.02) values were significantly higher in normobaric hypoxia compared to normoxia. These results suggest a parasympathetic dominance during acute exposure to normobaric hypoxia in PVD.

## 1. Introduction

Pulmonary vascular diseases (PVDs) including pulmonary arterial hypertension (PAH) and chronic thromboembolic pulmonary hypertension (CTEPH) are defined by an increased mean pulmonary artery pressure and pulmonary vascular resistance while the pulmonary artery wedge pressure (PAWP) as surrogate of left ventricular filling pressure is within normal limits [1]. PAH and CTEPH are defined as Group I and Group IV, according to the latest PH classification. Pulmonary vascular diseases [1] lead to right ventricular dysfunction with low cardiac output with progressive exertional dyspnoea as cardinal symptom and premature death [2,3].

The autonomic nervous system regulates the heart rate (HR) through sympathetic and parasympathetic discharging. Heart rate variability (HRV) is a non-invasive marker to assess cardiac autonomic modulation by fluctuations in consecutive R–R intervals [4]. It is necessary to distinguish between the interpretation of resting HRV (usually examined in 5 min intervals), 24 h HRV and HRV during sleep. HRV is a strong predictor for mortality in several disorders, such as acute myocardial infarction [5,6], heart failure [7] or sepsis [8].

Autonomic cardiovascular function has been studied in patients with PAH and impaired baroreflex function and increased sympathetic activity has been described [9,10,11,12]. However, the role of autonomic cardiovascular disturbance in progression of PAH or right ventricular dysfunction is not fully understood. Most of the previous studies detected increased sympathetic tone in patients with PAH [11,13]. A lower resting HR is associated with improved prognosis in PVD [14]. Furthermore, the activity of the sympathetic nervous system correlated with HR, oxygen saturation, New York Heart Association (NYHA) class, 6 min walk distance (6MWD) and pulmonary arterial flow acceleration time [15]. HRV has been shown to be a good predictor for disease severity and patient outcomes in subjects who suffer from PAH [2].

Both clinical conditions (e.g., worsening heart failure with hypervolemia, lower respiratory tract infections, intermittent acute pneumonia) and traveling by airplane and staying at altitude because of professional or recreational activities are exposing patients with PAH to acute or worsening hypoxia. The adaptive response to acute hypoxia in healthy individuals includes an increase in HR and cardiac output [16], an increase in respiratory rate and ventilation [17], and immediate hypoxic pulmonary vasoconstriction [18]. Thus, this hypoxic environment may induce adverse health effects in pre-existing PVD with an accelerated rise in PAP. Moreover, an increase in cardiac output might not be possible in patients with severe PAH. However, it is not exactly known whether short altitude sojourns or flights are potentially harmful to patients with PAH and how the autonomic cardiovascular regulation adapts.

The aim of this analysis of a randomized controlled blinded crossover trial was to investigate sympathovagal balance in stable patients with PAH/CTEPH exposed to short-term normobaric hypoxia using established measures of HRV.

## 2. Materials and Methods

### 2.1. Study Design, Intervention and Randomisation

In a randomized, controlled, single-blind crossover trial, participants were studied in random order on either ambient air or normobaric hypoxic air via a facemask attached to a two-way nonrebreathing valve (Hans Rudolph, Kansas City, MO, USA). The gas mixtures were provided by AltiTrainer^®^ (SMTEC-SA, Nyon, Switzerland) using ambient air (FiO_2_ = 0.21, normoxia) and nitrogen-enriched air (FiO_2_ = 0.15, normobaric hypoxia at 2500 m altitude) [19]. There was a >2 h wash-out period between the measurements.

All procedures performed were in accordance with the ethical standards of the national research committee and with the 1964 Helsinki Declaration and its later amendments. The study protocol has been approved by the Ethics Committee of the Canton of Zurich (KEK 2018-00455) and all research was performed in accordance with relevant regulations. The study protocol was registered at ClinicalTrials.gov (registration number NCT03592927).

### 2.2. Participants

Adult (age 18–75 years) patients with the diagnosis of PAH/CTEPH were recruited in this study from the outpatient Pulmonary Hypertension Unit of the University Hospital Zurich. PAH/CTEPH was diagnosed according to ESC/ERS guidelines [20]. Patients living at altitude <1000 m, having resting PaO_2_ ≥ 7.3 kPa, and stable condition on the same therapy for >4 weeks were eligible. Exclusion criteria were altitude exposure to >1500 m for ≥3 nights during the previous 4 weeks, long-term oxygen therapy with resting PaO_2_ ≤ 7.3 kPa, daytime hypercapnia with pCO_2_ > 6.5 kPa, relevant comorbidities (e.g., chronic obstructive pulmonary disease, heart failure, atrial fibrillation or flutter), treatment with betablocker, pregnancy and breast-feeding [19].

### 2.3. Assessments and Outcomes

Measurements were carried out in a standardised environment (in a quiet room with stable temperature and lightening) according to a strict protocol. Patients were exposed to either normoxia or normobaric hypoxia for 30–70 min before resting heart rate was measured for 20 min.

This study is a secondary analysis using data collected in a randomized controlled crossover trial that were not previously used [19]. Physiological parameters were assessed using Alice PDx (Philips Respironics, Murrysville, PA, USA), providing continuous three-lead electrocardiography (ECG) data at 1000 Hz, arterial oxygen saturation (SpO_2_), and breathing rate. Arterial blood gas analysis was performed at rest to provide information on partial pressure of O2 (paO_2_) and CO_2_ (paCO_2_) and arterial oxygen saturation (SaO_2_).

HRV was analysed in time- and frequency-domain (LabChart Software, ADInstruments Ltd., Sydney, Australia) in 5 min and 10 min traces as suggested by the European Society of Cardiology [21]. The minimum number of heartbeats available for analysis in one segment had to be >300 beats. The traces were screened, and sequences with movement artefacts or ectopic beats (<500 ms, >1200 ms) excluded manually. Time-domain parameters included the following: average HR, standard deviation of average HR (SDHR), average of RR intervals (mean RR), median of RR intervals (median RR), standard deviation of mean RR intervals (SDRR), coefficient of variation of RR intervals (CVRR), root mean squared sum difference of RR intervals (RMSSD) and the RR50 count divided by the total number of all RR intervals (pRR50). Frequency-domain parameters included the following: low frequency power (LF), high frequency power (HF) and the ratio of LF/HF. The HF, or so-called respiratory band (0.15–0.40 Hz), is influenced by breathing and represents the cardiac vagal control, while the LF band is influenced by both parasympathetic and sympathetic nervous systems [22].

The main outcomes of interest of this analysis were differences in the change of RMSSD, pRR50, LF and HF parameters of HRV between normoxia and normobaric hypoxia in patients with PVD. Secondary outcomes were the differences in other time- and frequency-domain parameters of HRV between normoxia and normobaric hypoxia in patients with PVD.

### 2.4. Statistical Analysis

Statistical analyses were performed with IBM SPSS (SPSS Inc., Chicago, IL, USA) and Graph Pad Prism 5.0 (GraphPad Software, San Diego, CA, USA). Normality of the data was evaluated using Shapiro–Wilk test and Kolmogorov–Smirnov test. Data were expressed as median with interquartile range (IQR; 25th–75th percentile). Changes of the variables between normoxia and normobaric hypoxia were assessed by ANOVA for repeated measurements with Fisher post hoc analysis or by using the Friedman test (nonparametric alternative to the one-way ANOVA with repeated measures) with Conover’s post hoc pairwise comparisons according to normality. A *p* value < 0.05 was considered statistically significant.

## 3. Results

### 3.1. Patients’ Characteristics

Of the total, 17/28 participants included in the original trial on the effects of normobaric hypoxia on exercise performance who finished the study per protocol had good quality resting ECG data, allowing HRV analysis in at least two 5–10 min sequences (Figure 1). Ten patients had to be excluded from the analysis because of ECG artefacts (four on normoxia, six on hypoxia), ectopic beats (present on either hypoxia or normoxia, no significant difference), or missing ECG data. The characteristics of the 17 patients are shown in Table 1.

### 3.2. HRV Analysis (Main Outcomes of Interest)

Significantly higher RMSSD (33.5 (27.1) vs. 20.8 (25.2) ms; *p* < 0.01) and pRR50 (2.8 (7.8) vs. 2.2 (3.4) ms; *p* = 0.03) values were detected in normobaric hypoxia compared to ambient air (Table 2, Figure 2). In addition to higher HRV indices in the time domain, significantly higher LF (558.6 (746.1) vs. 203.9 (425.6) ms^2^; *p* = 0.02) and HF (431.40 (661.6) vs. 183.7 (251.3) ms^2^; *p* < 0.01) values were found in normobaric hypoxia compared to ambient air (Figure 2). However, the LF/HF ratio did not change significantly under hypoxia (*p* = 0.14; Table 2).

### 3.3. Subgroup Analysis of HRV to Compare PAH and CTEPH and to Compare the Different Drug Treatment Groups

In the subgroup with idiopathic PAH (*n* = 8), the same statistically significant changes were found as in the total study population, and the direction of the effect of normobaric hypoxemia on the time- and frequency-domain parameters of HRV was also the same in CTEPH (*n* = 5) as in the total group, although in the small group of five patients, the majority of the results were not statistically significant. In a further subgroup, analysis comparing the results between the groups with endothelin receptor antagonist, phosphodiesterase inhibitor or combination therapy, the same direction of effect was found for all parameters as in the total group, although a statistically significant effect could no longer be demonstrated for all parameters.

### 3.4. HR Measures and Other HRV Data (Secondary Outcomes of Interest)

Exposure to normobaric hypoxia resulted in significantly higher SDRR (51.6 (28.7) vs. 36.3 (13.5) ms; *p* = 0.02) and CVRR (0.07 (0.02) vs. 0.05 (0.02) ms; *p* = 0.02) compared to ambient air. SDHR tended to be higher under hypoxia (3.8 (0.9) vs. 3.7 (2.1) beats/min; *p* = 0.06). HR itself and mean and median RR did not differ between the two conditions (all *p* > 0.05, Table 3). PaO_2_ values were significantly lower in normobaric hypoxia compared to ambient air (8.1 (1.2) vs. 10.7 (2.7) kPa; *p* < 0.01). paCO2 did not differ between the two conditions (all *p* > 0.05).

## 4. Discussion

In this randomized, controlled crossover trial, we investigated the effects of acute exposure to normobaric hypoxia compared to ambient air in patients with PVD (either PAH or distal CTEPH) in order to assess the autonomic cardiovascular adaptive response to hypoxia by different HRV measures. We found an increase in HRV in both the time-domain and the time–frequency-domain under normobaric hypoxia, while HR was unchanged. Although expecting rather a reduction in HRV due to sympathetic activation, we found an increase in HRV and particularly measures such as RMSSD and pRR50 (both representing mainly the parasympathetic arm and high-frequency variations) as well as LF and HF in short-term exposure to normobaric hypoxia. Our results suggest parasympathetic dominance whilst patients were resting quietly and breathing under normobaric hypoxia compared to ambient air according to a randomized, single-blinded crossover design.

HRV analysis is a simple method to assess the status of the autonomic nervous system and to predict cardiovascular morbidity and mortality [23]. The RMSSD parameter represents parasympathetic activity [23] and it is the simplest to assess because of its stability [23,24]. The LF measures represent the efferent activity of parasympathetic and sympathetic activity. The HF measures represent only the efferent parasympathetic activity [25]. Short-term recordings of HRV are preferred in clinical practice. The frequency domain methods should be performed in 5 min segments [23]. Time domain methods, especially RMSSD, can also be used to evaluate short-term recordings [23]. In line with these recommendations, we also analysed 5–10 min segments.

Patients with PAH seem to have a lower HRV than healthy subjects [2]. Previous studies in healthy participants suggested that acute hypoxia results in a decrease in HRV and parasympathetic tone and increased sympathetic activity [26]. In line with this, most of the studies demonstrated decreased HF and LF values of HRV at high-altitude [27,28,29,30]. In contrast to our study, most of these studies were performed at real altitude, exposing study participants, mostly healthy individuals, to hypobaric hypoxia. Inconsistent findings have been reported on the effect of hypoxia on RMSSD in healthy subjects in nonrandomised and uncontrolled studies: a significant decrease in RMSSD was detected in some [26,31], but not in all studies [32,33]. A previous study in healthy individuals proposed 6000 m (FiO_2_ = 9.8%) of simulated altitude as threshold to induce changes in resting cardiac autonomic modulation [34]. Our study, assessing resting HRV in PVD, simulated an altitude of 2500 m with an FiO_2_ = 0.15 (normobaric hypoxia). Acute exposure to hypoxia resulted in decreased resting HRV and sympathetic dominance and withdrawal of vagal control in healthy volunteers [35]. However, our data of short-term exposure of PAH/CTEPH to normobaric hypoxia demonstrated increased HRV and parasympathetic dominance. Of interest, a blunted response to hypoxia in PVD patients was previously described by Carta A et al. [36]. This study showed, in 149 PVD-patients assessed by right heart catheterization, that the mPAP did not relevantly change under hypoxia, but significantly decreased whilst breathing pure oxygen, pointing towards a blunted hypoxic pulmonary vasoconstriction. In line with this, Groth et al. [37] showed that an increase in mPAP in response to breathing hypoxic gas is significantly higher in healthy groups compared to patients with PVD. In the present study, we found an increase in HRV while HR itself was unchanged. However, if RMSSD and pRR50 quantify parasympathetic modulation driven by ventilation, a theoretical change in ventilation during exposure to hypoxia might affect HRV by ventilatory modulation of R–R intervals. Vagal activity is the major contributor to the HF component, a marker that consistently and significantly increased in our study population in response to hypoxia. Although the difference was not significant, the LF/HF ratio tended to be lower in hypoxia and a low LF/HF ratio also reflects parasympathetic dominance. Of interest, however, repetitive exposure of healthy individuals to normobaric hypoxia over 10 days (1 h on each day) also resulted in increased HRV and HRV measures of parasympathetic dominance [38] indicating that hypoxic preconditioning might play a role.

PAH/CTEPH is characterised by sympathetic activation and consequential reduced resting HRV [39,40]. Hypoxia as stressor could result in a further increase in sympathetic activity. Interestingly, our results suggest rather a parasympathetic activation under short-time normobaric hypoxia in PAH/CTEPH. A lack of hypoxia-induced sympathetic overactivity is also reflected by the absence of an increase in HR, breath-rate, and pulmonary artery pressure in this cohort with PVD [19].

Commercial aircraft and altitude sojourns could be dangerous for patients with PVD because of hypobaric hypoxia. Exposure to hypoxia could lead to exaggerated hypoxic pulmonary vasoconstriction with an increase in pulmonary vascular resistance in patients with PVD resulting in altitude-related adverse health effects (ARAHE) [41]. However, according to studies by right heart catheterization, hypoxic pulmonary vasoconstriction in a normobaric-hypoxic setting seems to be blunted in PVD [36]. A high-altitude simulation test is an alternative way of exploring altitude-related changes, but its predictive value seems to be limited [41,42]. There are limited data on cardiovascular effects of a short-term altitude stay in patients with PAH. A previous study by Schneider et al. [43] investigated the differences in blood oxygenation and echocardiographically assessed PAP between simulated altitude (normobaric hypoxia; FiO_2_ = 15%~2500 m) and real altitude (hypobaric hypoxia; 2500 m, Mount Saentis). Patients with PAH/CTEPH presented more pronounced hypoxemia and pulmonary hemodynamic changes during the short-term exposure to the real altitude compared to simulated altitude. Thus, the authors concluded that the use of high-altitude simulation test is questionable to predict the cardiorespiratory changes of patients with PAH/CTEPH at altitude or during air travel [43]. Other studies support the fact that the predictive value of the high-altitude-simulating test for ARAHE is limited, which has also been shown in a recent study in 75 COPD-patients traveling to 3100 m [41,42,44]. Therefore, resting HRV in PAH/CTEPH needs to be assessed during hypobaric hypoxia at real altitude and during a longer exposure to hypoxia as a next step.

A limitation of our study is the small number of patients. However, PVD are exceedingly rare and the randomized crossover design allows to considerably reduce the sample size needed to demonstrate an effect. Because the present study examined the short-term effects of normobaric hypoxia, it does not allow conclusions to be drawn about the effects of hypobaric hypoxia during acute or longer-term altitude exposure in PVD. To better understand the adaptive changes of HRV in PVD during exposure to real altitude, further studies should be performed. Another limitation is the lack of comparability with the effect of the same exposure to normobaric hypoxia in healthy subjects, since only patients with PAH were studied. However, a randomized controlled trial offers many advantages over a case–control study. Nevertheless, healthy subjects should be studied under normobaric hypoxia in the future to better understand the effect on HRV. Another limitation is that the original study was powered to detect estimated minimally important difference in constant work-rate exercise test (CWRET) and not in HRV measures [19]. However, the fact that 11/28 patients could not be included in the per protocol analysis because of the insufficient quality of the ECG measurements or many ectopic beats may introduce a selection bias.

## 5. Conclusions

This randomized, controlled crossover trial found an increase in HRV in patients with PAH/CTEPH exposed to short-term normobaric hypoxia, suggesting parasympathetic activation. Thus, it does not support the finding of an increase in sympathetic activity described in healthy individuals exposed to hypobaric hypoxia.

## Figures and Tables

**Figure 1 jcm-12-01782-f001:**
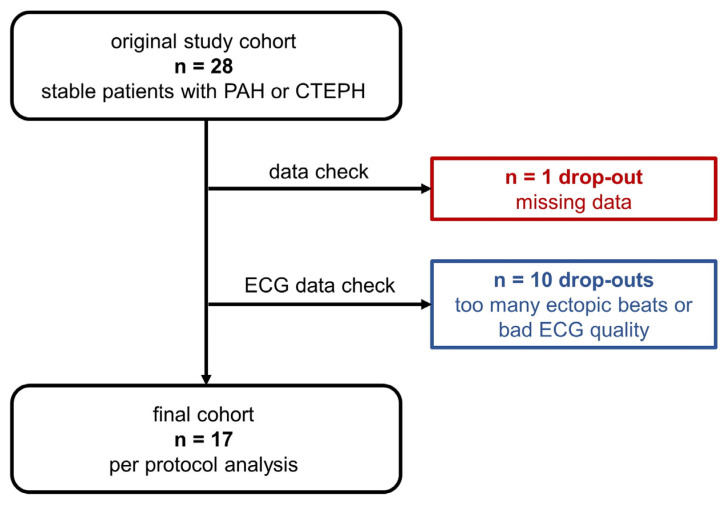
Patients’ flow.

**Figure 2 jcm-12-01782-f002:**
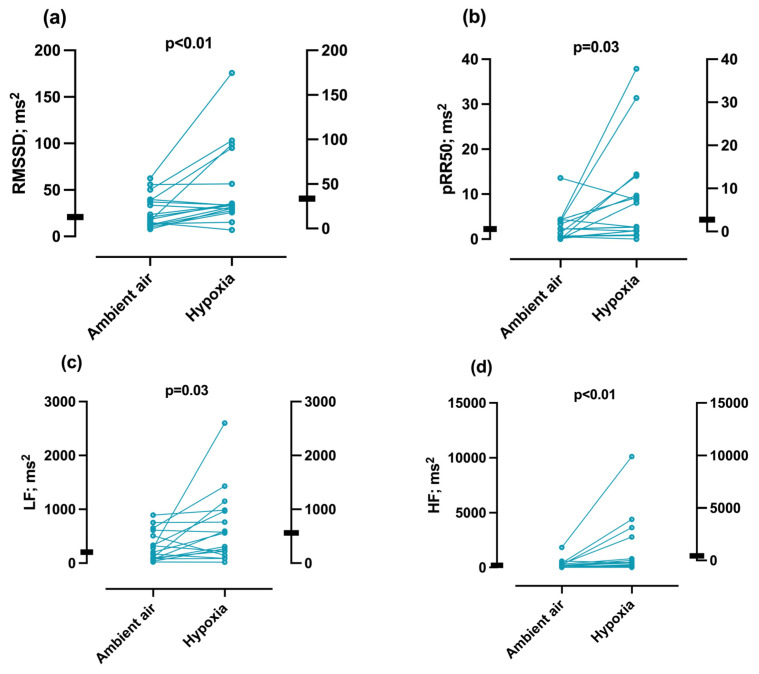
Difference in HRV parameters between ambient air and normobaric hypoxia. (**a**) RMSSD (ms). (**b**) pRR50 (ms). (**c**) LF (ms). (**d**) HF (ms). Each point shows a measurement of a patient. The black rectangles represent the group median. HF—high-frequency band; LF—low-frequency band; LF/HF—ratio of LF/HF; pRR50—percentage of differences higher than 50 ms in R–R intervals; RMSSD—square root of the sum of all differences between successive R–R intervals.

**Table 1 jcm-12-01782-t001:** Patients’ characteristics.

N	17
Men; *n* (%)	6 (35)
Age; years	63.0 (11.0)
BMI; kg/m^2^	24.0 (4.4)
Pulmonary hypertension classification; *n* (%)	
1. Pulmonary arterial hypertension	12 (71)
1.1. Idiopathic	8 (47)
1.4.1. Connective tissue disease	3 (18)
1.4.3. Portal hypertension	1 (6)
4. Distal chronic thromboembolic pulmonary hypertension	5 (29)
New York Heart Association functional class; *n* (%)	
I	5 (29)
II	8 (47)
III	4 (24)
Targeted therapy; *n* (%)	
Endothelin receptor antagonist	13 (76)
Phosphodiesterase-5 inhibitor	9 (53)
Soluble guanylate cyclase stimulator	0 (0)
Prostacyclin- or receptor agonist	2 (12)
Combination	8 (47)
Resting PaO_2_ at 490 m; kPa	10.8 (2.8)
Resting SpO_2_ at 490 m; %	95.9 (2.9)
Resting PaCO_2_ at 490 m; kPa	4.4 (0.5)
Pulmonary arterial pressure; mmHg	43.0 (23.0)
Pulmonary artery wedge pressure; mmHg	11.5 (2.3)
Pulmonary vascular resistance; Wood units	5.4 (5.0)
Resting cardiac output; L/min	4.7 (1.0)
Time spent under normoxia; min	53 (16)
Time spent under hypoxia; min	57 (7)

Data are presented as median with IQR (25th–75th percentile).

**Table 2 jcm-12-01782-t002:** Differences in main HRV parameters between ambient air and normobaric hypoxia.

*n* = 17		Ambient Air	Normobaric Hypoxia	*p*
Time domain	RMSSD; ms	20.8 (25.2)	33.49 (27.1)	<0.01
	pRR50; ms	2.2 (3.4)	2.75 (7.8)	0.03
Frequency domain	LF; ms^2^	203.9 (425.6)	558.60 (746.1)	0.02
	HF; ms^2^	183.7 (251.3)	431.40 (661.6)	<0.01
	LF/HF; %	3.2 (23.5)	2.12 (2.1)	0.14

Data are presented as median with IQR (25th–75th percentile). HF—high-frequency band; LF—low-frequency band; LF/HF—ratio of LF/HF; pRR50—percentage of differences higher than 50 ms in R–R intervals; RMSSD—square root of the sum of all differences between successive R–R intervals.

**Table 3 jcm-12-01782-t003:** Other HRV parameters, HR and blood gas parameters in ambient air and normobaric hypoxia.

*n* = 17	Ambient Air	Normobaric Hypoxia	*p*
Total beats analysed; *n*	439.2 (111.7)	460.3 (160.01)	0.71
Average RR; ms	830.6 (99.9)	840.9 (99.5)	0.75
Median RR; ms	832.1 (101.2)	842.1 (102.8)	0.75
SDRR; ms	36.3 (13.5)	51.6 (28.7)	0.02
CVRR; ms	0.05 (0.02)	0.07 (0.02)	0.02
Average HR; beats/min	73.0 (9.1)	73.6 (9.2)	0.85
SDHR; beats/min	3.7 (2.1)	3.8 (0.9)	0.06
PaO_2_; kPa	10.7 (2.7)	8.1 (1.2)	<0.01
SpO_2_; %	95.8 (2.3)	92.4 (3.7)	0.07
PaCO_2_ in Zurich; kPa	4.9 (0.5)	4.9 (0.5)	0.76
pH	7.5 (0.04)	7.5 (0.03)	0.33

Data are presented in median with IQR (25th–75th percentile). CV—coefficient of variance of R–R intervals; HR—heart rate; RR—R–R intervals; SDHR—standard deviation of average HR; SDRR—standard deviation of R–R intervals.

## Data Availability

The data are available from the corresponding author on request.

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
