# Peer review of "Effects of Acute Hypoxia on Heart Rate Variability in Patients with Pulmonary Vascular Disease"

_jcm, 2023, doi:10.3390/jcm12051782_

Round 1

Reviewer 1 Report

This was a secondary analysis from a single blinded crossover trial looking at individuals with pulmonary arterial hypertension, either Group 1 or group 4, and heart rate variability measures in relation to normobaric hypoxia.  Individuals with pulmonary arterial hypertension who were not already acclimatized to high-altitude were exposed to both normoxia nd hypoxia for reportedly more than 20 minutes and multiple indices of heart rate variability were measured.  Individuals were exposed to both normoxia and hypoxia in randomized fashion with at least 2 hours between sessions .  The authors found that there were significant differences with the hypoxemic sessions in which multiple measures of heart rate variability, the root mean square difference of the RR and the pRR50, counts were elevated compared to normoxic conditions.  Additionally, both high-frequency and low-frequency measurements were higher in the exposure to hypoxia.  The authors concluded that in individuals with pulmonary arterial hypertension exposed to a normobaric hypoxic environment exhibited a greater parasympathetic activation during this acute exposure.  This finding was different from previous findings in healthy individuals as well as others with pulmonary hypertension, though those previous studies were mostly performed in hypobaric hypoxic conditions. 

Major critique:

Probably the most significant weakness of the study design is the lack of healthy controls.  One strength of the study is that individuals with PH recruited to the study underwent both hypoxic and normoxic measurements (serving as their own controls), it would be nice to have contextualized these results by comparison to other healthy individuals without pulmonary hypertension.  Particularly since the authors found that the parasympathetic response was greater with a higher rate of derivative measures of heart rate variability, which they note is different findings compared to previous studies in healthy individuals as well as those with pulmonary arterial hypertension who underwent a different design in which they were exposed both hyperbaric and hypoxemic conditions.  Lack of healthy controls here does not invalidate what they found, but having healthy individuals as part of the study would have strengthened their discussion and conclusion.  I certainly do not think this precludes the study results from publication, but I think further discussing lack of controls both in the discussion and as a limitation to the study is warranted.

Minor critique:

In the introduction on line 47, "24–HRV" is not defined.  I assume that it means 24 hour heart rate variability, but not every reader may make that assumption and it should be clearly defined.

There are multiple times throughout the paper in which the authors use "healthy" as a noun without the preceding definitive article "the."  I knew what they were talking about, but I am not aware of English grammar in which healthy is used as a noun without a definitive article. “ Healthy” can also be used as an adjective.  For example, they could say "healthy" individuals, or "the" healthy, but I have never seen "healthy" commonly used as a noun without a preceding definitive article.  This happens throughout the introduction and discussion section.  Would recommend checking and correcting.

The authors allude to this being a secondary analysis in the discussion section, but it is not clear initially in the method section that this was actually part of the previously published study.  I would respectfully request that this be described a little bit more carefully within the method section.

The authors note in the assessment and outcomes section that the patients receive either normoxia or hypoxia for greater than 20 minutes.  However, it is nowhere listed exactly how long each patient experienced specific levels of oxygen concentrations each time.  What does greater than 20 minutes really mean?  Did they get 20 minutes exactly? 30 minutes? Did someone get an hour?  Would they please be more specific about how much time the subjects actually spent in each category and list this in table 1?

Line 111–“artifact” is misspelled.  Please correct

In table 1, they list “pulmonary arterial pressure”.  The legend mentions that the data is presented as median.  What is the “pulmonary artery pressure” actually representing?  Is this the mean pulmonary arterial pressure or systolic pulmonary arterial pressure?  I suspect that the authors are presenting the median of the mean pulmonary artery pressure, but could they please confirm and make this more clear in table 1?

I was able to follow the discussion points through the discussion, but the prose format would benefit from further evaluation and some improvement from a grammatical standpoint.  For example, the very first line in discussion section reads as a run-on sentence.  The authors would benefit by making their points by separating the sentence into 2 or 3 points.  This happens a couple of times throughout the discussion section. 

On the line 222 there is a typo where the PRR50 is written as "PNN50"

Author Response

PLEASE SEE ATTACHMENT in case prefered as PDF 

Thank you for your valuable comments and questions and for the opportunity to revise our manuscript. We have commented on all questions and comments on the following pages and revised the manuscript accordingly. We hope that our manuscript in its current form can be considered for publication in the Journal of Clinical Medicine.

Sincerely,

Martina Meszaros and Esther Irene Schwarz

Reviewer 1

This was a secondary analysis from a single blinded crossover trial looking at individuals with pulmonary arterial hypertension, either Group 1 or group 4, and heart rate variability measures in relation to normobaric hypoxia. Individuals with pulmonary arterial hypertension who were not already acclimatized to high-altitude were exposed to both normoxia and hypoxia for reportedly more than 20 minutes and multiple indices of heart rate variability were measured.  Individuals were exposed to both normoxia and hypoxia in randomized fashion with at least 2 hours between sessions. The authors found that there were significant differences with the hypoxemic sessions in which multiple measures of heart rate variability, the root mean square difference of the RR and the pRR50, counts were elevated compared to normoxic conditions. Additionally, both high-frequency and low-frequency measurements were higher in the exposure to hypoxia. The authors concluded that in individuals with pulmonary arterial hypertension exposed to a normobaric hypoxic environment exhibited a greater parasympathetic activation during this acute exposure. This finding was different from previous findings in healthy individuals as well as others with pulmonary hypertension, though those previous studies were mostly performed in hypobaric hypoxic conditions.

Question 1: Probably the most significant weakness of the study design is the lack of healthy controls.  One strength of the study is that individuals with PH recruited to the study underwent both hypoxic and normoxic measurements (serving as their own controls), it would be nice to have contextualized these results by comparison to other healthy individuals without pulmonary hypertension. Particularly since the authors found that the parasympathetic response was greater with a higher rate of derivative measures of heart rate variability, which they note is different findings compared to previous studies in healthy individuals as well as those with pulmonary arterial hypertension who underwent a different design in which they were exposed both hyperbaric and hypoxemic conditions. Lack of healthy controls here does not invalidate what they found but having healthy individuals as part of the study would have strengthened their discussion and conclusion. I certainly do not think this precludes the study results from publication, but I think further discussing lack of controls both in the discussion and as a limitation to the study is warranted.

Response: Thank you for the thoughts on possible differences in the effect of normobaric hypoxia between patients with PAH and healthy individuals, which would be really interesting to investigate. However, this is a randomized controlled cross-over study in patients with PAH and unfortunately there are no data on healthy individuals that would allow a group comparison. The fact that the data are from an RCT, in our opinion, makes the results stronger, especially compared to a case-control study in PAH and healthy subjects, and the cross-over design gives greater statistical power, but we discuss in the limitations paragraph that unfortunately the interpretation compared to healthy subjects is missing, which would have been of great interest for the physiological understanding of our results. Future studies are warranted recruiting healthy controls to understand our findings.

Question 2: In the introduction on line 47, "24–HRV" is not defined. I assume that it means 24 hour heart rate variability, but not every reader may make that assumption and it should be clearly defined.

Response: Thank you for comment. We have corrected this word according to its meaning, as 24-hour HRV measurement. 

Question 3: There are multiple times throughout the paper in which the authors use "healthy" as a noun without the preceding definitive article "the."  I knew what they were talking about, but I am not aware of English grammar in which healthy is used as a noun without a definitive article. “ Healthy” can also be used as an adjective.  For example, they could say "healthy" individuals, or "the" healthy, but I have never seen "healthy" commonly used as a noun without a preceding definitive article.  This happens throughout the introduction and discussion section.  Would recommend checking and correcting.?

Response: Thank you for the suggestion for improvement, which we gladly accept. In the revised manuscript, either “the healthy” is used as a noun or *healthy” as an adjective.

Question 4: The authors allude to this being a secondary analysis in the discussion section, but it is not clear initially in the method section that this was actually part of the previously published study.  I would respectfully request that this be described a little bit more carefully within the method section.

Response: Thank you for the suggestion specifying the type of analysis. We describe this analysis of data from a randomized controlled trial as a secondary analysis because previously collected, previously unused data are used for analysis in a new study (definition secondary analysis). ECG data and indices of HRV measurements were not used in the original study by Schneider et al.

Question 5: The authors note in the assessment and outcomes section that the patients receive either normoxia or hypoxia for greater than 20 minutes. However, it is nowhere listed exactly how long each patient experienced specific levels of oxygen concentrations each time. What does greater than 20 minutes really mean? Did they get 20 minutes exactly? 30 minutes? Did someone get an hour?  Would they please be more specific about how much time the subjects actually spent in each category and list this in table 1?

Response: Thank you for this important advice. We included the exact time frame (defined as 30-70 minutes) under normoxia and hypoxia in the Methods section, at the end of which approximately 20 minutes of resting ECG was examined. We also included the variable "time under normoxia and hypoxia" in Table 1 (data are given as median with IQR (25th-75th percentile).

Time spent under normoxia; min

53 (16)

Time spent under hypoxia; min

57 (7)

Question 6: Line 111–“artifact” is misspelled.  Please correct.

Response: Thank you for your note. We have corrected the misspelled word.

Question 7: I was able to follow the discussion points through the discussion, but the prose format would benefit from further evaluation and some improvement from a grammatical standpoint. For example, the very first line in discussion section reads as a run-on sentence. The authors would benefit by making their points by separating the sentence into 2 or 3 points. This happens a couple of times throughout the discussion section.

Response: Thank you for the suggestion to make shorter sentences. We have rewritten this part of the manuscript, especially separating the sentences that were too long. We hope that the revised sentences will be easier to understand this way.

Question 8: On the line 222 there is a typo where the PRR50 is written as "PNN50"

Response: Thank you, we corrected the misspelled word.  

Reviewer 2 Report

This was an interesting article. Figure 2 demonstrates some participants had an increase in HRV whereas others did not. Were there particular factors that differentiated who did vs who did not? For example, is this associated with the particular aetiology of PVD, or the treatment the patient was receiving? Subgroup analysis by PVD cause (e.g. Connective tissue disease, CTEPH) and by treatment (e.g. endothelin receptor antagonist) should be performed.

Also, it would be wise to repeat exactly the same experiment using control ie. healthy subjects, and compare the results.

Author Response

PLEASE SEE ATTACHMENT in case prefered as PDF 

Thank you for your valuable comments and questions and for the opportunity to revise our manuscript. We have commented on all questions and comments on the following pages and revised the manuscript accordingly. We hope that our manuscript in its current form can be considered for publication in the Journal of Clinical Medicine.

Sincerely,

Martina Meszaros and Esther Irene Schwarz

Reviewer 2

Question 1: This was an interesting article. Figure 2 demonstrates some participants had an increase in HRV whereas others did not. Were there particular factors that differentiated who did vs who did not? For example, is this associated with the particular aetiology of PVD, or the treatment the patient was receiving? Subgroup analysis by PVD cause (e.g. Connective tissue disease, CTEPH) and by treatment (e.g. endothelin receptor antagonist) should be performed.

Response: Thank you for raising this important point. We performed the following analyses:

(1) subanalysis among patients with (a) idiopathic pulmonary arterial hypertension (n = 8) and (b) with CTEPH (p = 5). 2nd subanalysis of patients with (a) endothelin receptor antagonist therapy (n = 13), (b) phosphodiesterase 5 inhibitor therapy (n = 9), and (c) combination therapy (n = 8). The other subgroups were not analyzed because of the small number of participants. Our results can be found below. These findings will be mentioned in the results section and discussed, but due to the small number in the subgroups and the therefore hardly justifiable statistical comparisons and p-values, not listed in detail in the manuscript. The findings were identical in PAH compared to the whole group and the direction of the effect was the same in CTEPH (but not statistically significant in the analysis with only 5 CTEPH patients). Also for all three medication subgroups, the effect direction of exposure to normobaric hypoxia on time domain and frequency domain HRV measures was identical to the finding in the whole study population, but in low sample sizes of subgroups not always statistically significant. In summary, subgroup analyses did not identify a specific group with a different response.

Patients with idiopathic pulmonary hypertension

n=8

Ambient air

Normobaric hypoxia

p

Time domain

RMSSD; ms

17.64 (15.72)

33.57 (4.79)

0.039

pRR50; ms

0.87 (2.18)

5.38 (8.84)

0.023

Frequency domain

LF; ms2

87.74 (263.86)

288.65 (463.84)

0.055

HF; ms2

159.75 (152.84)

339.10 (313.19)

0.039

LF/HF; %

12.921 (9.82)

1.52 (0.35)

0.023

Patients with CTEPH

n=5

Ambient air

Normobaric hypoxia

p

Time domain

RMSSD; ms

17.84 (20.35)

32.66 (60.35)

0.094

pRR50; ms

1.46 (2.85)

1.86 (23.39)

0.219

Frequency domain

LF; ms2

243.08 (350.15)

565.78 (1009.56)

0.313

HF; ms2

120.89 (232.13)

257.53 (2126.01)

0.094

LF/HF; %

13.01 (16.26)

1.35 (4.59)

0.063

Patients with endothelin receptor antagonist therapy

n=13

Ambient air

Normobaric hypoxia

p

Time domain

RMSSD; ms

23.65 (26.32)

33.82 (64.93)

0.002

pRR50; ms

2.25 (3.96)

8.07 (7.06)

0.021

Frequency domain

LF; ms2

282.25 (509.12)

572.95 (761.70)

0.057

HF; ms2

183.70 (267.00)

431.400 (560.11)

0.006

LF/HF; %

14.24 (12.81)

1.26 (0.84)

0.305

Patients with phosphodiesterase-5 inhibitor therapy

n=9

Ambient air

Normobaric hypoxia

p

Time domain

RMSSD; ms

20.76 (20.00)

33.49 (4.64)

0.074

pRR50; ms

2.24 (3.02)

2.75 (6.12)

0.203

Frequency domain

LF; ms2

203.90 (430.96)

308.05 (446.10)

0.570

HF; ms2

163.70 (127.53)

407.35 (306.60)

0.098

LF/HF; %

14.24 (11.97)

1.55 (0.51)

0.012

Patients with combined therapy

n=8

Ambient air

Normobaric hypoxia

p

Time domain

RMSSD; ms

22.20 (18.83)

33.65 (9.62)

0.078

pRR50; ms

2.25 (3.34)

5.41 (6.32)

0.250

Frequency domain

LF; ms2

263.98 (493.32)

433.33 (432.94)

0.461

HF; ms2

173.70 (93.62)

419.38 (213.73)

0.109

LF/HF; %

16.09 (12.03)

1.60 (0.60)

0.023

Question 2: Also, it would be wise to repeat exactly the same experiment using control ie. healthy subjects, and compare the results.

Response: Thank you for your comment. Since this was a randomised cross-over trial in patients with PAH/CTEPH, we cannot provide data on healthy people, but we agree that studying the effect of normobaric hypoxia in healthy using the same protocol would be interesting and helpful for the interpretation of our findings. However, compared to a case control study or matching PAH patients with healthy, a randomised controlled cross-over design has several benefits. Please see also answer to comment 1 of reviewer 1. However, it is an important point, and it has been discussed in the Limitation section of the manuscript.

Round 2

Reviewer 2 Report

-